# Dietary Habits and Race Day Strategies among Flexitarian, Vegetarian, and Vegan Recreational Endurance Runners: A Cross-Sectional Investigation from The NURMI Study (Step 2)

**DOI:** 10.3390/nu16111647

**Published:** 2024-05-27

**Authors:** Derrick R. Tanous, Mohamad Motevalli, Claus Leitzmann, Gerold Wirnitzer, Thomas Rosemann, Beat Knechtle, Katharina Wirnitzer

**Affiliations:** 1Department of Sport Science, University of Innsbruck, 6020 Innsbruck, Austria; 2Department of Secondary Education, University College of Teacher Education Tyrol, 6010 Innsbruck, Austria; 3Institute of Nutrition, University of Gießen, 35390 Gießen, Germany; 4adventureV & change2V, 6135 Stans, Austria; 5Institute of Primary Care, University of Zurich, 8006 Zurich, Switzerland; 6Medbase St. Gallen, Am Vadianplatz, 9000 St. Gallen, Switzerland; 7Research Center Medical Humanities, University of Innsbruck, 6020 Innsbruck, Austria; 8Department of Pediatric Oncology and Hematology, Charité—Universitätsmedizin Berlin, 13353 Berlin, Germany

**Keywords:** plant-based, omnivore, marathon, athlete, training, sport, macronutrients, protein

## Abstract

Many of today’s recreational runners have changed their diet from omnivorous to vegetarian or vegan for reasons like better sport performance, animal ethics, positive health, eco-aspects, or male infertility. Others have constructed the flexitarian diet due to current trends in sustainable eating. The aim of this investigation was to analyze the dietary habits and race day strategies of recreational endurance runners following current sustainable dietary trends. Recreational endurance runners (18+ years) were invited to complete the standardized online survey on socio-demography/anthropometry, motivations, running/racing history, food frequency, and race day dietary strategy. Chi-squared tests and Wilcoxon tests were used for the statistical analysis. In total, 289 participants submitted the survey; 146 subjects following flexitarian (*n* = 34), vegetarian (*n* = 50), or vegan (*n* = 62) diets were included in the final sample. Significant differences were found across the diet types: BMI (*p* = 0.018), fruit/vegetable consumption (*p* < 0.001), and the dietary motive of performance (*p* = 0.045). The findings suggest that the flexitarian diet may be appropriate for health- and environmentally conscious populations living in a meat-centered society and lacking social support to eat completely vegetarian/vegan. Following a plant-based diet is perceived as easy for health-conscious, athletic populations, and the vegan diet does not require a particularly effortful/complex race day strategy for endurance runners.

## 1. Introduction

Long-distance racing (e.g., half-marathon, marathon) is a highly popular sport across Europe, with millions of performers worldwide [1]. Endurance runners are known to be a healthy population based on reports of regular exercise behavior, health-conscious dietary habits, and little to no alcohol or substance abuse [2,3,4]. Likewise, a higher prevalence of vegetarian and vegan diets has been found among endurance runners compared to the general population [4,5,6]. On race day, runners often deviate from their usual dietary and fluid intake [7], possibly due to physical and psychological stress [8], fundamental motivations [9], and an individualized race day strategy [10]. 

Vegetarian (including vegan) runners mainly appear motivated to adhere to their diet type for ethical reasons [9,11], which may limit their race day dietary intake fluctuations regardless of health. Flexitarians, on the other hand, are more lenient with dietary practices [12]. Robust definitions for the classification of vegetarian and vegan diet types have been reported by the Academy of Nutrition and Dietetics [13]. A stringent definition from leading nutritional organizations for flexitarian is lacking, however. Similar to the lack of scientific consensus on the term “plant-based diet”, the most colloquially or socioculturally accepted explanation of the flexitarian diet is usually following a vegetarian pattern but occasionally consuming meat [12,14,15,16,17]. It has been reported that flexitarians have similar underlying dietary motivations as vegetarians, including for health or reducing one’s carbon footprint [18]. The major dietary motivational difference between flexitarians and vegetarians appears to be the heightened concern among vegetarians for animal welfare [11,18]. Across studies, one dilemma associated with defining the flexitarian diet is the connection between diet type adherence and time [18,19,20]. 

Exceptional health benefits have been found among people following plant-based diets compared to the typical omnivorous diet [21], especially regarding the direction of the effect for diets that are proportionally more and more plant-based (flexitarian to vegetarian to vegan) [22]. Concerns regarding certain nutrients have been proposed for people following any general type of diet [23], but especially for diets that exclude specific food categories, such as meat or even all animal products [13]. Although there are often critical reports of people following the vegan diet, including but not limited to protein deficiency [24], vitamin B12 inadequacy [25], and even severe malnutrition [26], populations eating vegan diets actually appear to have much greater control over their health in general and especially in the long term [13,21,22,27]. Among athletes, however, there is still a general lack of evidence regarding the special strategies necessary for people following more plant-based diets when safely and effectively competing in sports [28,29,30,31,32]. The male athlete competing in sports is renowned for meat and protein consumption [30,33] and domination across all life aspects, which has been considered synonymous with masculinity and even sexual potency [34,35,36] until the realization of sound scientific evidence available today [37]. Still, among underinformed sports nutrition experts, the answer to the question “Where do plant-based (especially vegan) athletes get their protein?” remains elusive [38].

The NURMI study was developed to investigate the depth of the general diet types (omnivore, vegetarian, vegan) in the sport of recreational endurance running, as limited scientific evidence has been published in the area [39] and the health-related evidence of plant-based nutrition suggests that further examination in special populations is needed [28,32]. The aim of the present investigation was to compare the dietary habits and race day strategies of recreational endurance runners following current dietary trends in sustainable eating (flexitarian, vegetarian, and vegan diets). It was assumed that recreational endurance runners following vegan diets require a complex race day dietary strategy compared to flexitarians or vegetarians to meet the demands of long-distance running performance and optimal recovery.

## 2. Materials and Methods

The NURMI (Nutrition and Running High Mileage) study was conducted with a cross-sectional design and based on recreational endurance runners. The ethics committee of St. Gallen, Switzerland, approved the study protocol (May 2015; EKSG 14/145) [40], a trial registry was performed (ISRCTN73074080), and the STROBE-nut reporting guidelines were followed [41]. The NURMI study was planned out in 3 steps, where Step 2 of the NURMI study included the most detailed questionnaire and was prepared to analyze comprehensive variables of running and lifestyle, with a special focus on plant-based aspects. For more details on Step 2, please refer to the previous publications [2,9,42,43,44,45,46,47,48,49,50,51,52,53]. 

### 2.1. Study Procedure

The study participants were endurance-running adults, mainly from Austria, Germany, and Switzerland. Several channels were used as a base for study recruitment, such as social media, personal contacts, marathon race websites, webpages of running communities, and email subscribers to running magazines, including health and wellness, lifestyle, nutrition, trade fairs on sport topics, and plant-based diets. Before study participation was possible, the subjects were briefed on the study procedure in writing and the participants were required to provide informed consent. The questionnaire was available online [54] over the course of 11 months (February–December 2015) in English or German. The survey questions were focused on physical and psychological health topics and included a basic classification of the individual’s running purpose. Whether for well-being, hobby, or competition, the running purpose was linked with the subjects’ running, racing, and training motivation and additional physical activities that were performed parallel to running. 

To reach complete participation in the NURMI study Step 2, it was required to fulfill all the inclusion criteria: (1) submission of the written informed consent document, (2) being 18 years of age (at least), (3) submission of the completed Step 2 questionnaire, (4) and having finished a half-marathon race (or greater distance) over the course of the prior two years. In addition, to be included in the present investigation, it was required to (5) self-report following a vegetarian or vegan diet. 

### 2.2. Subjects

A total of 57 subjects preferred to run the 10 km distance and were included as a further comparator subgroup due to the high-quality data that were contiguous to the half-marathon and marathon runners. Endurance runners typically compete over increasingly longer distances with the accumulation of experience in racing. Therefore, runners who preferred the ultra-marathon distance were excluded from the present investigation due to having an upper level or extreme refinement in terms of their race day strategy. For Step 3 of the NURMI study, the subjects were required to select a long-distance running event to plan for the “NURMI running event” [40].

To verify the subjects’ reliability with regard to the survey answers, specific topics had control questions spread across various sections, such as running participation (motivations, competition and experience, training aspects, etc.) as well as for the dietary classification. Regarding the validated food frequency questionnaire (FFQ) that was used and the definition of the 53 food groups/clusters of the “German Health Interview and Examination Survey for Adults” (DEGS-FFQ), the detailed methodology was previously described in detail [9,48,49], as well as for supplement intake [45,46,47]. In order to classify the runners into the three refined dietary subgroups (flexitarian, vegetarian, vegan), a three-step control process was followed. Initial self-reports of vegetarian or vegan diets were controlled with the race day dietary strategy, and runners consuming meat for racing were classified as flexitarian (e.g., “race day flexitarian”). The secondary control included the shifting of vegetarian runners who reported eating vegetarian on race day but initially self-reported as vegan. The tertiary level control was based on the runners’ food frequency reports over the previous four weeks and followed the leading definitions of the diet types: vegetarian (no meat/fish but possible dairy or egg products) and vegan (no consumption of animal sources products, such as processed or non-processed meat, fish/seafood/shellfish, eggs, dairy products, or honey) [13,55]. In total, the control analysis revealed that 45 runners had to be shifted to other dietary categories: 24 from vegetarian to flexitarian (16% of the total sample) and 21 from vegan (10 to flexitarian and 11 to vegetarian; 14% of the total sample).

The subjects’ datasets were excluded from the study if contradictory or conflicting answers were identified or if important questions were not answered. Subsequently, a Body Mass Index (BMI) procedure was implemented in this study to exclude subjects with a BMI greater than or equal to 30 kg/m^2^, pertaining to the appropriate health recommendation of the World Health Organization (WHO) [56,57]. Therefore, the BMI procedure was implemented to control for a minimum level of health among the subjects. Likewise, reaching a safe body weight with a weight-management strategy is a prerequisite before distance-running participation and is associated with additional health promotion. In Figure 1, the subjects’ flow of enrolment and dietary subgroup classification is shown. The subjects’ characteristics are available in Table 1. 

### 2.3. Statistical Analysis

The runners’ race day strategies were outlined considering specific variables connected to the diet type: motivation (leisure, performance), training guidance (independent, professional, other), prevalence of race day sport supplement use and the reason (muscle gain, performance, energy/nutrients, recovery, replenish, regeneration, health, other), the nutrient and liquid intake strategy (same as always, same as for training, different on race day, by feeling), total fluid intake and type (water, isotonic sports drink, other), in-race caloric intake (kcal), and in-race macronutrient intake (carbohydrates, protein, fat).

The statistical analysis was completed with R Core Team (version 4.2.2 UCRT) [58]. The exploratory analysis was conducted using descriptive statistics, including means with standard deviations (SDs) as well as medians with ranges. Non-parametric tests were used to analyze the significant differences between dietary subgroups considering racing experience and history. Chi-square tests (χ^2^; nominal scale) and Wilcoxon tests (ordinal/metric scales) were used to confirm the associations between variables. Food frequency clusters were defined using 53 manifest variables (considering the frequency and amount of the specified foods). A heuristic index (compound variable) with a range from 0 to 100 was defined (equal for all the items), and the FFQ was calculated by multiplying the two questions and dividing by the maximum (in order to scale the dietary intake by measures, items, and clusters); a simple linear regression model was used to estimate the means and 95% confidence intervals (the values have been used in Figure 2). A binomial regression model was used to estimate the influences of diet and training on body weight loss. Box plots were used to display the differences in the race day macronutrient intake by the dietary subgroups (flexitarian, vegetarian, vegan). The level of statistical significance was set at *p* ≤ 0.05.

## 3. Results

Overall, the questionnaire was completed by 289 participants. The final sample included a total of 146 subjects, as 143 participants did not meet all of the criteria for inclusion. The subjects were mainly from Germany and Austria (*n* = 134; 92%), were predominantly female (*n* = 97; 66%), married (*n* = 60; 88%), and had an average age of 38.5 years (SD ± 10.5). In terms of academic backgrounds, most subjects held a university degree (*n* = 54; 37%) or had completed high school (*n* = 44; 30%). Table 1 displays the sociodemographic backgrounds of the endurance runners considering the diet type.

Across the diet-type subgroups, a significant difference was found for the BMI (*p* = 0.018), with flexitarians having the highest (23 kg/m^2^, range: 18–26). No significant differences were found for the body weight (*p* = 0.191) or height (*p* = 0.643) across the subgroups. For the racing preferences, most subjects preferred to run the half-marathon distance (*n* = 57; 39%) or 10 km distance (*n* = 57; 39%), with no significant difference between the subgroups (*p* = 0.272). The subjects had comparable health and leisure motivations at the start of running across the diet types (*p* = 0.079); a significant difference was found between the diet types considering the motivation to race (*p* = 0.046), with the vegetarians (*n* = 31; 67%) and vegans (*n* = 32; 55%) being mostly performance-motivated and the flexitarians mostly motivated for leisure (*n* = 19; 61%).

For the racing experience, a significant difference was found (*p* = 0.003), where flexitarians had the most experience at 10 years (SD ± 9), and the vegans had the least at 5 years (SD ± 5). For the racing history, no significant differences were found across the diet-type subgroups (*p* > 0.05); the subjects had an average age of 31 years (SD ± 10) at their first race, 33 years (SD ± 10) at their first half-marathon, 35 years (SD ± 9) at their first marathon, and had completed a total of 11 races (SD ± 12).

Table 2 displays the dietary motivation and experience of the distance runners by the diet-type subgroups. A significant difference was found for the diet-type adherence motive of performance (*p* = 0.045), where the vegan subgroup was the most motivated to follow their current diet type considering performance purposes (*n* = 36; 60%). Most of the runners had changed their diet type to the current diet accordingly (*n* = 124; 85%); a significant difference was found in the proportions of the diet-type subgroups regarding a stated diet-type change (*p* = 0.003), where vegans were the least likely to have followed their diet type across the lifespan (*n* = 2; 3%). Of the subjects who had changed their diet type, the most commonly reported previous diet type followed was omnivorous (*n* = 63; 51%), with a significant difference across the subgroups (*p* = 0.001), where most vegans reported following a vegetarian diet previously (*n* = 37; 62%).

Figure 2 displays the food frequency consumption of the diet-type subgroups, which corresponds to the amounts of each food cluster and statistical analysis provided in Appendix A. Figure 3 displays the body weight fluctuations (gone down, stable, gone up) concerning the dietary subgroups (flexitarian, vegetarian, vegan). No significant differences were found between the dietary subgroups considering the weight changes due to diet (*p* = 0.822) or training separately (*p* = 0.652). The vegans had the most stability regarding the body weight fluctuations due to diet (16%; *n* = 5) or training (50%; *n* = 29). A binomial regression analysis showed a significant difference in the weight fluctuations between diet vs. training, where diet had a greater influence than training on the weight changes (b = –1.17; SE = 0.35; *p* = 0.001).

Table 3 displays the strategy for performing in long-distance running events by diet type. No significant difference was found for the training guidance (*p* = 0.556), where most subjects had no guidance (*n* = 106; 77%). The prevalence of sport supplement use on race day was comparable across the diet types (*p* = 0.053), with most participants not using any supplements on race day (*n* = 82; 63%). A significant difference was found for one (Energy/Nutrients) of the eight possible reasons for using a sport supplement (*p* = 0.037), where flexitarians were the most likely to use a supplement for this reason (*n* = 14; 88%) and vegans were the least likely (*n* = 11; 52%). No significant difference was found for the reports of nutrient and liquid intake on race day across the diet types (*p* = 0.609), and the most common report was that the runners have a specific nutrient and liquid intake strategy for race day (*n* = 52; 40%). No differences were found for the fluid intake (*p* > 0.05), whether water, an isotonic drink, or other, based on the diet-type subgroups as well as the caloric intake (*p* = 0.536) or macronutrient intake proportions of carbohydrates, fat, and protein (*p* > 0.05). Figure 4 displays the comparison of the macronutrient caloric consumption by diet type on race day.

## 4. Discussion

The NURMI study was established as the first of its kind to compare long-distance runners around the world following omnivorous, vegetarian, and vegan diets. This investigation aimed to analyze the dietary habits of recreational endurance runners following flexitarian, vegetarian, and vegan diets and to compare their race day strategies. The most important results include: (i) flexitarians had a significantly greater BMI than vegetarians and vegans; (ii) diet had a significantly greater effect on body weight loss than long-distance running training; (iii) flexitarians were consuming the most animal products; (iv) vegans had the highest protein intake; (v) most of the subjects have changed their diet type since childhood; (vi) diet-type changes were mostly considered to be easy; (vii) 9 out of 10 dietary motives were similar across the subgroups, with the exception of performance in sports, where vegans demonstrated the most motivation; and (viii), as the main result of the present investigation, the race day strategies were similar across the subgroups.

As the NURMI study was developed as an exploratory investigation to research potential diet-type differences in the area of sports and recreational long-distance running [40], the subjects displayed an elevated level of health status and health consciousness as compared to the general population [2,43,44], which appears to go hand-in-hand with reduced meat consumption [14,18,19,21,22,23,27,30,59,60,61]. This aspect was also reflected in the present investigation, with a special focus on plant-based athletes, where the BMI values were consistently found to be in the normal range across the diet-type subgroups [56]. However, a significant difference was identified across the subgroups, where the flexitarians displayed the highest BMI. This finding is consistent with previous reports [22,59], as large cohorts analyzing diet-type differences have found that people eating greater proportions of animal products, even just white meat and fish, have higher BMIs (significantly greater overweight and obesity prevalence) than people eating proportionally more plant-based foods, with what appears to be a linear relationship, and further analysis by future systematic review and meta-regression is suggested. It should be mentioned that the primary mechanism for body weight status is personal dietary behavior [32,60,61,62,63], and this is further regulated by physical activity levels as a weight management strategy [64,65,66], which is consistent with the diet vs. training binomial regression results of the present investigation. This finding, depicted in Figure 3, offers a basic introduction to body weight control and management for individuals interested in a healthy and sustainable approach to reducing their body weight [60,61,62,63,64]. Likewise, the implications of body weight fluctuations are especially relevant for distance runners, who may need to pay careful attention to maintaining a stable body weight over stretches of intensive training and racing [43,50,52]. The BMI calculation as a single component, however, is an insufficient indicator of health and does not take body composition into account, such as muscle, adipose, or bone tissue [56,67], although the BMI has been shown to be an accurate indicator of body fatness with a large study sample [68].

In addition, when comparing the BMI values of the subgroups with the food frequency reports in the present investigation, it is shown that the flexitarians were eating the most animal products, including dairy, meat and fish, eggs, and animal protein, with significant differences across the subgroups (*p* < 0.001). Likewise, the vegans displayed the most fundamentally healthy dietary behavior considering the consumption of whole plant foods, including legumes, nuts, and pulses, and fruit and vegetables, with significant differences across the subgroups (*p* < 0.05). This finding is consistent with the level of plant protein intake reported by the vegan subgroup, which has been shown to significantly reduce the risk of cardiovascular disease, all-cause mortality, and even male infertility when substituting for many varieties of animal protein sources [36,69,70]. Interestingly, it was found that the vegans (e.g., the people consuming only plant-based foods) had the highest protein intake overall, with a significant difference across the diet types. As protein is a type of macronutrient, which is essential for the body to grow and sustain life, it must be regularly consumed when on any type of diet [71]. Even among nutrition experts, it has been reported that a considerable degree of confusion exists regarding protein intake, especially for athletes [29,30]. For example, the term “high-quality protein” is used to indicate protein sources containing the complete amino acid profile (e.g., contains all the essential amino acids that the body cannot produce and must be regularly consumed) and is related to foods that are almost exclusively of animal origin (except for some plant foods like quinoa, soybean, or flaxseed among others) [72,73]. The term “high-quality protein” is therefore misleading, especially for experts [73,74,75], as “high-quality protein” does not refer to the functional quality of the end product (e.g., the quality of the protein as a functioning part of muscle tissue). Irrespective of including “high-quality protein” sources in the diet, plant-based athletes can easily meet the demands of protein intake by eating more calories in general and incorporating a wide variety of whole plant-based foods in the day-to-day diet (e.g., eating at least a minimum of legumes such as varieties of beans, peas, or lentils together with a whole grain such as barley, wheat, rice, corn, oat, etc.) [29,30]. Based on the food frequency reports, it appears that a healthy dietary consciousness was present among the vegans, although it has been reported as mandatory for people following a vegan diet to consume a B12 vitamin supplement (either 50 µg daily or 2000 µg weekly) [76] and to closely monitor the following micronutrients: vitamin D, omega-3, iron, iodine, and zinc [13]. However, attention to a critical variety of nutrients is also important for people following omnivorous or even flexitarian diets, such as fiber, low-density lipoprotein and total cholesterol, folate, sodium, and heme-iron [77,78,79].

Considering that the vast majority of the global population consumes some variety of the omnivorous diet [5,80], the present investigation on plant-based athletes uncovered that 85% (*n* = 124) of the subjects had changed their diets since childhood. In addition, although there were a majority of reports of dietary change across the subgroups, a significant difference showed that the vegans were the most likely to have changed their diets, at 97% (*n* = 60), mostly switching from previously consuming a vegetarian diet (62%; *n* = 37) but some had also switched directly from omnivorous nutrition (33%; *n* = 20). One possible reason that a high prevalence of dietary change has been reported among this health-conscious sample of recreational endurance runners may be due to the omnivore’s dilemma [81]. An individual’s attachment to meat consumption has been self-reported to be due to factors like hedonism, entitlement, and dependency, which are closely tied to social norms, the fallacy of human supremacy, and the make-up of one’s own dietary identity [82]. Likewise, given the conclusive scientific evidence regarding health deterioration from proportions of animal products as part of the diet [21,22,79,83,84,85,86] and the possibility of safely excluding all animal products [13], there may also be a fragile relationship between the personal diet and the environment [80]. Furthermore, it was found that a small percentage of the vegetarians had switched their diet from previously following a vegan diet (5%; *n* = 2). This finding may be linked to the addictive quality of cheese, considering the circumstances of casomorphins previously reported [87,88]. 

An extensive majority of the subjects (94%; *n* = 116) reported that it was easy to complete a dietary change, with no difference between the subgroups, even amongst the vegan subjects [89]. This finding may be difficult to conceptualize considering that the flexitarians had not realized a totally restrictive dietary change (e.g., meaning that they were still consuming any food category without personal sacrifice to appropriate a more sustainable eating pattern) [20], as opposed to the vegetarians or especially by the vegans [90], which may be why all of the flexitarians (*n* = 34) reported no challenge to change. On the other hand, this finding shows that switching to a vegetarian or vegan diet was perceived as easy for health-conscious individuals but may seem challenging for people who have no experience [89]. It was also reported by the majority of subjects that the previous dietary change affected their physical or mental well-being, with virtually all the subjects reporting a positive change. This finding is likely due to consuming more foods from whole plant sources than the previous diet, such as fruits, vegetables, whole grains, and legumes (as well as the synergistic effects of eating the varieties together), which is related to a healthier dietary profile of protein and fat (with little to no dietary cholesterol in the case of the vegan diet), folate, fiber and complex carbohydrates, phytochemicals, and especially antioxidants [78,91].

The motivation behind following a vegetarian or vegan diet is primarily ethical concerns regarding animal welfare [9,11]. According to the com-b model, however, a person’s capabilities and opportunities may also, in turn, reflect dietary behavior [92]. Given the strong societal foundation that is visible around the world concerning the belief that meat consumption is necessary for reproductivity and survival, strength, or even evolution [30,36,37,71,82], when a person refuses to consume meat, interpersonal disputes typically arise due to the phenomenon of cognitive dissonance (e.g., when two or more personal values conflict) [93]. Such interpersonal communication arising from the personal choice not to eat has been so powerful it has effectively been used to stop mass violence, for example [94]. For women with male partners, there may be a lack of courage to argue with their partner to eat only vegan and to follow what the more dominant partner (typically the male) wants on the plate [30,92]. With the situation of vegetarians, and even more so of vegans, the power of the animal welfare dietary motivation exceeds the social pressures (inclusive of subtle or open discrimination) of family, friends, and even teammates [11,30], and also in the face of social environmental constraints [92]. However, among flexitarians, the societal pressures appear to surpass one’s own desires to fully eat vegetarian or vegan, which may be related to having health or environmental reasons as a primary motive for this dietary pattern rather than animal ethics [18,93]. In the present investigation, which was based on a refined sample of healthy, active recreational distance runners, it was found that there was no difference between the dietary subgroups considering the dietary motivation for ethical reasons, although the pattern from previous investigations was still visible regarding vegans having the highest interest in ethical aspects, followed by the vegetarians [9,11,95]. Interestingly, in the case of this study, the runners following flexitarian diets were classified into their dietary subgroup based upon the desire to be vegan or vegetarian but contradictorily consuming meat for racing, which is likely related to the fact that the majority of people who follow vegetarian or vegan diets were raised on an omnivorous diet. As adults are rarely capable of terminating an unhealthy behavior once it has developed over childhood, adolescence, or even young adulthood [96,97], this would imply that these participants were struggling with the behavior change process, according to the transtheoretical model of behavior change [98,99], and were under considerable stress to perform on race day. The main result considering the runners’ dietary motivations showed that the vegans had the highest interest in following their diet for sports performance reasons, which is particularly noteworthy due to the lack of scientific evidence displaying a performance advantage by following a vegan diet when compared to other general diet types [53,100,101]. However, it has previously been suggested that possible anti-inflammatory benefits may exist from diets composed of higher proportions of plant foods, which could be a possible mechanism for enhancing endurance-related athletic performance [28]. Indeed, further research is needed to conclude sports performance differences based on diet types (such as long-term and/or lifetime vegetarian or vegan dietary adherence), especially considering before-and-after trials, as competitive athletes who follow a vegan diet often report this factor as the reason for their success [30].

Regarding the race day strategies, and especially the race day dietary strategies of the subgroups, it was found that there was no difference considering the means of training guidance by diet type. This result is consistent with previous findings [52,102] and most likely due to the fact that the subjects were recreational athletes and not professionals, and therefore, professional guidance is limited to the minority of subjects most probably highly motivated to perform [103]. For using a sport supplement on race day, no difference was found across the subgroups, as most subjects did not consume any supplement (63%; *n* = 82), which shows that it is viable to consume a meat-free diet without any additional supplementation to perform in long-distance, endurance running events. In line with this finding is the report that flexitarians were the most likely to use a sport supplement for energy or nutrients on race day, which may be related to the fact that they consumed meat under special circumstances, although their usual diet excluded meat. As the macronutrient energy supply of meat is primarily fat and/or protein, the bioavailability is rather limited for efficient and continuous energy [104], especially during racing events [30]. Although no differences were found considering the diet-type subgroups for macronutrient intake on race day, the flexitarians displayed the highest fat and protein intakes. In comparison, the vegans were the least likely to consume a sport supplement on race day as a functional food item to cope with the higher exercise—and especially—race-induced needs for energy or nutrients (mainly carbohydrates), which may be a distinct benefit of this diet for performance reasons. 

It was found that the nutrient and liquid intake planning strategy was similar across the subgroups, with the most frequent report being a different strategy for races, which is consistent with the previous research [105,106]. The subjects mostly consumed water or an isotonic sports drink during races, which has been found to enhance performance, with the carbohydrate content suggested to be between 6 and 8% as a standard for gastrointestinal acceptance and comfort [106]. For caloric intake on race day, most subjects consumed up to 3000 calories, although some consumed up to 4000 or even up to 6000 calories on race day. It has been recommended to meet caloric needs in order for optimal recovery following long-duration, endurance activity [107], and the longer the race distance, the more calories are required, which can be difficult to plan, especially for beginners in long-distance races. Therefore, the findings of the present investigation indicate that following a vegan diet can be easy and compatible for running and applying as a dietary strategy for performance in recreational endurance races. As no particularly complicated race day dietary strategy was observed for runners following a vegan diet, this investigation rejects the assumption that recreational endurance runners adhering to vegan diets require a complex race day dietary strategy compared to flexitarians or vegetarians. However, for recreational endurance runners in general, close attention is advised to maintaining a good health status, and consultation with up-to-date professionals in the field of sports nutrition may be beneficial for long-term dietary sustainability and performance.

The present investigation has limitations to consider along with the findings. The two primary limitations are due to the cross-sectional design and the self-reporting by the subjects. As the NURMI study was developed as the first of its kind to examine omnivorous, vegetarian, and vegan diet-type differences in the performance of long-distance races around the world, it was necessary to access a large sample, which was in favor of following the cross-sectional design with self-reporting. However, the self-reports were monitored considering control questions throughout the survey, which aided in excluding false information. In addition, the sample size of the present investigation was considerably small for the breadth of recreational endurance runners worldwide, but the three-step dietary control procedure helped to develop the refined dietary subgroups that were finally included and analyzed. The present sample was mostly married (*n* = 60; 88%) and had an average age of 38.5 years, which is contrary to common veggie profiles, who are often single/unmarried and younger (approx. 14–30 years) [30,108]. However, the subjects in this investigation were predominantly female (*n* = 97; 66%), which aligns with previous reports [11]. Likewise, participation in the NUMRI study was voluntary, which may limit the results to the highly motivated recreational endurance runners. However, to the best of our knowledge, this is the first study to analyze flexitarians in endurance running sports and the new findings and interesting indications underline the need for more research and future studies.

## 5. Conclusions

This study is the first to compare the race day strategies of flexitarian, vegetarian, and vegan recreational endurance runners. The flexitarians in the present study desired to consider themselves vegetarian or vegan, given current trends in sustainable eating. However, potentially due to the lack of sociocultural support to eat completely veggie or vegan—considering the high pressure of the meat-centered society around them (competitors, family, the circle of friends, etc.)—the flexitarian diet therefore appears to be a golden path for certain individuals to find their comfort zone, with occasional meat consumption but also following the desire to eat mostly to completely vegetarian/vegan. The findings show that following a plant-based diet is perceived to be easy for health-conscious populations and that following the vegan diet does not require a particularly effortful or complex race day strategy for endurance runners. Therefore, the results of the present investigation may be quite useful for endurance runners and their trainers or coaches, as well as for experts in physiology, sports and exercise, (sports) nutrition and dietology, or medicine.

## Figures and Tables

**Figure 1 nutrients-16-01647-f001:**
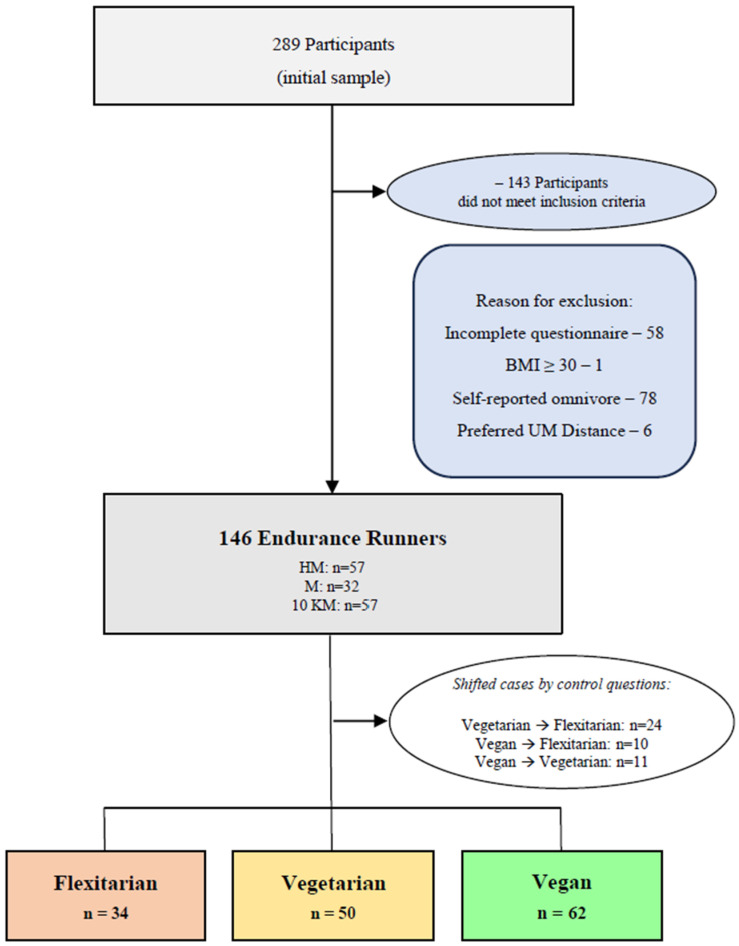
Enrolment in the study flowchart, including dietary categories. BMI—body mass index. HM—half-marathon; M—marathon; 10 KM—10 km.

**Figure 2 nutrients-16-01647-f002:**
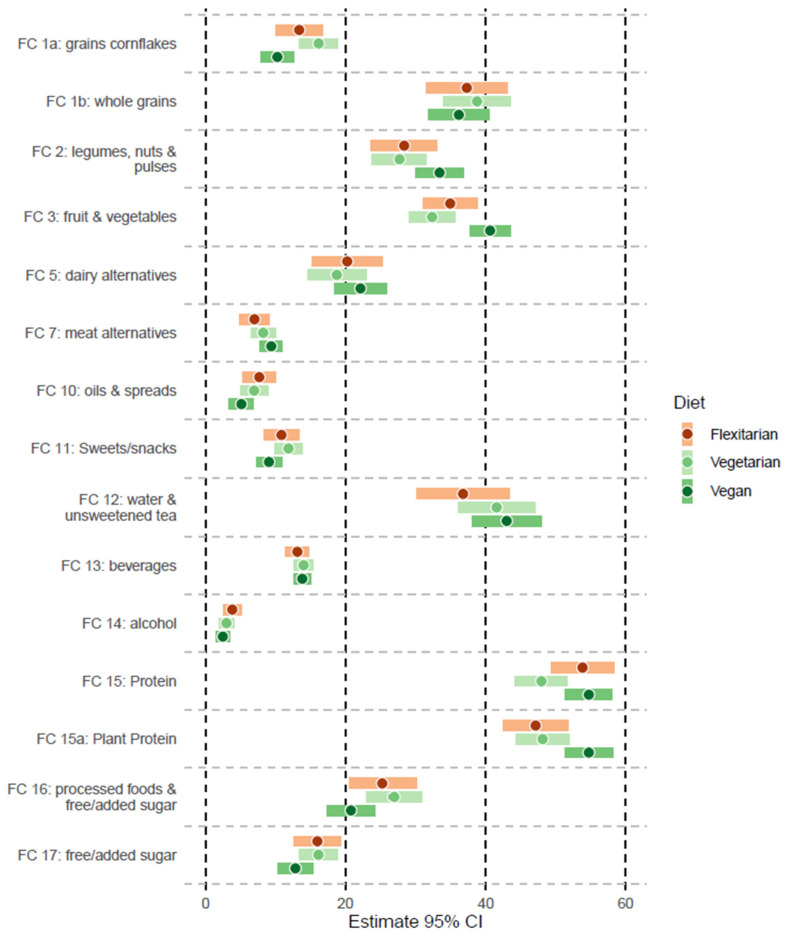
Food frequency consumption across diet-type subgroups displayed by forest plot with 95% confidence interval.

**Figure 3 nutrients-16-01647-f003:**
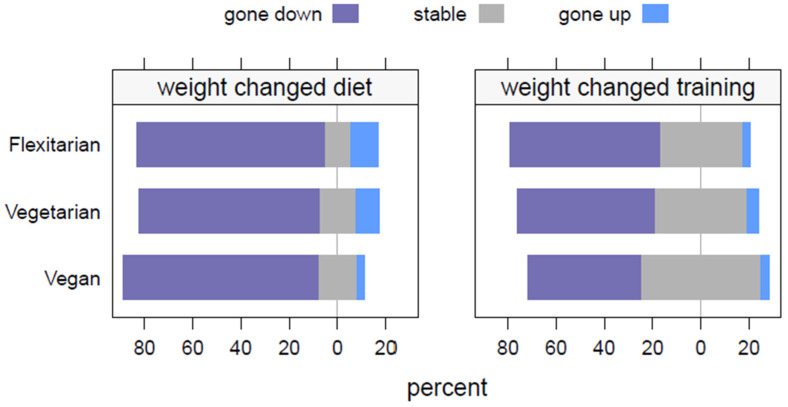
Body weight fluctuations across dietary subgroups: diet vs. training.

**Figure 4 nutrients-16-01647-f004:**
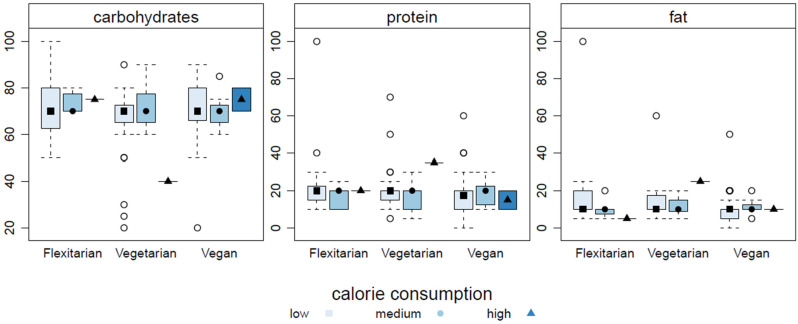
Box plots of race day macronutrient intake by dietary subgroups (flexitarian, vegetarian, vegan).

**Table 1 nutrients-16-01647-t001:** Sociodemographic backgrounds of endurance runners by diet-type subgroups.

		Total	Flexitarian	Vegetarian	Vegan	Statistics
		100% (146)	23% (34)	34% (50)	43% (62)	
**Nationality**	Germany Austria Switzerland Other	77% (113) 14% (21) 3% (4) 5% (8)	79% (27) 21% (7) / /	80% (40) 10% (5) 4% (2) 6% (3)	74% (46) 15% (9) 3% (2) 8% (5)	χ^2^_(6)_ = 5.64 *p* = 0.465
**Sex**	Female Male	66% (97) 34% (49)	56% (19) 44% (15)	62% (31) 38% (19)	76% (47) 24% (15)	χ^2^_(2)_ = 4.58 *p* = 0.101
**Age** **(years)**		38.5 ± 10.5	41.4 ± 11.3	38.5 ± 10.6	37 ± 9.89	F_(2, 143)_ = 1.85 *p* = 0.161
**Body Weight (kg)**		63.2 ± 9.59	65.9 ± 10.4	61.8 ± 9.34	62.9 ± 9.17	F_(2, 143)_ = 1.68 *p* = 0.191
**Height (m)**		1.71 ± 0.09	1.71 ± 0.08	1.71 ± 0.09	1.70 ± 0.09	F_(2, 143)_ = 0.44 *p* = 0.643
**BMI** **(kg/m^2^)**		21.0 (17–28)	23.0 (18–26)	21.0 (17–26)	21.0 (17–28)	F_(2, 143)_ = 4.13 *p* = 0.018
**Academic Qualification**	None N/A A Level Upper Secondary University Degree	<1% (1) 12% (17) 21% (30) 30% (44) 37% (54)	/ 12% (4) 26% (9) 32% (11) 29% (10)	/ 12% (6) 20% (10) 30% (15) 38% (19)	2% (1) 11% (7) 18% (11) 29% (18) 40% (25)	χ^2^_(8)_ = 3.00 *p* = 0.934
**Marital Status**	Single Married Divorced	32% (46) 88% (60) 8% (12)	29% (10) 65% (22) 6% (2)	36% (18) 58% (29) 6% (3)	29% (18) 60% (37) 11% (7)	χ^2^_(4)_ = 1.88 *p* = 0.757
**Main Race Distance**	10 km HM M	39% (57) 39% (57) 22% (32)	29% (10) 38% (13) 32% (11)	36% (18) 46% (23) 18% (9)	47% (29) 34% (21) 19% (12)	χ^2^_(4)_ = 5.15 *p* = 0.272
**Initial Motivation to Run**	Leisure Health	56% (82) 44% (64)	44% (15) 56% (19)	68% (34) 32% (16)	53% (33) 47% (29)	χ^2^_(2)_ = 5.07 *p* = 0.079
**Motivation to Race**	Leisure Performance	44% (60) 56% (75)	61% (19) 39% (12)	33% (15) 67% (31)	45% (26) 55% (32)	χ^2^_(2)_ = 6.18 *p* = 0.046
**Racing Experience (years)**		7 ± 7	10 ± 9	8 ± 6	5 ± 5	F_(2, 142)_ = 6.18 *p* = 0.003
**Age at First Race (years)**		31 ± 10	33 ± 11	30 ± 10	31 ± 10	F_(2, 142)_ = 0.82 *p* = 0.442
**Age at First HM (years)**		33 ± 10	35 ± 10	32 ± 11	32 ± 9	F_(2, 126)_ = 0.63 *p* = 0.536
**Age at First M (years)**		35 ± 9	36 ± 9	34 ± 11	34 ± 8	F_(2, 74)_ = 0.23 *p* = 0.791
**Total Races Finished**		11 ± 12	9 ± 8	13 ± 12	10 ± 14	F_(2, 143)_ = 1.76 *p* = 0.176
**HM Races Finished**		2.67 ± 2.66	2.50 ± 2.54	2.74 ± 2.38	2.71 ± 2.97	F_(2, 143)_ = 0.50 *p* = 0.607
**M Races Finished**		1.29 ± 2.59	1.71 ± 2.78	1.06 ± 1.87	1.24 ± 2.97	F_(2, 143)_ = 1.89 *p* = 0.155

Note. Results are presented in percentages (%), total numbers, mean (SD), and median (range). χ^2^ statistic calculated by Pearson’s chi-square test and F statistic by Kruskal–Wallis test. kg–kilograms. m–meters. N/A–no answer. 10 km—10 km. HM—half-marathon. M—marathon.

**Table 2 nutrients-16-01647-t002:** Dietary motivation and experience by diet-type subgroups.

		Total	Flexitarian	Vegetarian	Vegan	Statistics
		100% (146)	23% (34)	34% (50)	43% (62)	
**Diet Motives**	Health Ethics Environment World Hunger Performance Taste Food Scandals Economics Religion Tradition	88% (109) 85% (106) 80% (99) 56% (70) 50% (62) 41% (51) 35% (44) 19% (23) 6% (8) 2% (3)	92% (24) 73% (19) 77% (20) 50% (13) 50% (13) 46% (12) 27% (7) 15% (4) 8% (2) 8% (2)	84% (32) 84% (32) 79% (30) 61% (23) 34% (13) 34% (13) 47% (18) 18% (7) 8% (3) /	88% (53) 92% (55) 82% (49) 57% (34) 60% (36) 43% (26) 32% (19) 20% (12) 5% (3) 2% (1)	χ^2^_(2)_ = 0.97; *p* = 0.615 χ^2^_(2)_ = 5.12; *p* = 0.077 χ^2^_(2)_ = 0.28; *p* = 0.869 χ^2^_(2)_ = 0.70; *p* = 0.705 χ^2^_(2)_ = 6.19; *p* = 0.045 χ^2^_(2)_ = 1.14; *p* = 0.565 χ^2^_(2)_ = 3.56; *p* = 0.169 χ^2^_(2)_ = 0.26; *p* = 0.880 χ^2^_(2)_ = 0.41; *p* = 0.816 χ^2^_(2)_ = 4.15; *p* = 0.126
**Diet Changed**	Yes No	85% (124) 15% (22)	76% (26) 24% (8)	76% (38) 24% (12)	97% (60) 3% (2)	χ^2^_(2)_ = 11.81 *p* = 0.003
**Previous Diet**	Omnivorous Vegetarian Vegan	51% (63) 45% (56) 4% (5)	81% (21) 19% (5) /	58% (22) 37% (14) 5% (2)	33% (20) 62% (37) 5% (3)	χ^2^_(4)_ = 17.98 *p* = 0.001
**Duration of Previous Diet (years)**		20.0 (0.2–57.5)	32.4 (0.6–57.5)	19.6 (0.6–40.5)	15.1 (0.2–53.0)	F_(2, 121)_ = 6.24 *p* = 0.003
**Difficulty of Diet Change**	Easy Challenging	94% (116) 6% (8)	100% (26) /	95% (36) 5% (2)	90% (54) 10% (6)	χ^2^_(2)_ = 3.13 *p* = 0.209
**Diet Change Affected Physical/Mental Well-being**	Yes No	79% (98) 21% (26)	88% (23) 12% (3)	61% (23) 39% (15)	87% (52) 13% (8)	χ^2^_(2)_ = 11.36 *p* = 0.003
**Physical/Mental Well-being Effect**	Positively Negatively	99% (97) 1% (1)	100% (23) /	96% (22) 4% (1)	100% (52) /	χ^2^_(2)_ = 3.29 *p* = 0.193

Note. Results are presented in percentages (%), total numbers, and median (range). χ^2^ statistic calculated by Pearson’s chi-square test and F statistic by Kruskal–Wallis test.

**Table 3 nutrients-16-01647-t003:** Race strategy by diet-type subgroups.

		Total	Flexitarian	Vegetarian	Vegan	Statistics
		100% (146)	23% (34)	34% (50)	43% (62)	
**Training Guidance**	None Professional Other	77% (106) 15% (20) 8% (11)	73% (24) 18% (6) 9% (3)	72% (33) 17% (8) 11% (5)	84% (49) 10% (6) 5% (3)	χ^2^_(4)_ = 3.01 *p* = 0.556
**Sport Supplement Use** **on Race Day**	Yes No	37% (49) 63% (82)	55% (16) 45% (13)	27% (12) 73% (32)	36% (21) 64% (37)	χ^2^_(2)_ = 5.87 *p* = 0.053
**Reason for Sport Supplement Use**	Gain Muscle Performance Energy/Nutrients Recovery Regeneration Replenish Health Other	2% (1) 39% (19) 71% (35) 12% (6) 31% (15) 57% (28) 4% (2) 4% (2)	/ 38% (6) 88% (14) 12% (2) 50% (8) 69% (11) 6% (1) /	/ 33% (4) 83% (10) 8% (1) 8% (1) 58% (7) / 8% (1)	5% (1) 43% (9) 52% (11) 14% (3) 29% (6) 48% (10) 5% (1) 5% (1)	χ^2^_(2)_ = 1.36; *p* = 0.506 χ^2^_(2)_ = 0.31; *p* = 0.857 χ^2^_(2)_ = 6.59; *p* = 0.037 χ^2^_(2)_ = 0.25; *p* = 0.881 χ^2^_(2)_ = 5.68; *p* = 0.059 χ^2^_(2)_ = 1.66; *p* = 0.435 χ^2^_(2)_ = 0.73; *p* = 0.695 χ^2^_(2)_ = 1.26; *p* = 0.533
**Nutrient & Liquid Intake** **on Race Day**	Same as Always Same as Training Different for Races By feeling	5% (7) 33% (43) 40% (52) 22% (29)	7% (2) 21% (6) 41% (12) 31% (9)	5% (2) 34% (15) 36% (16) 25% (11)	5% (3) 38% (22) 41% (24) 16% (9)	χ^2^_(6)_ = 4.50 *p* = 0.609
**Fluid Intake in Race (L)**	Water Isotonic Drink Other	0.68 ± 0.65 0.22 ± 0.36 0.09 ± 0.29	0.65 ± 0.52 0.25 ± 0.28 0.07 ± 0.19	0.66 ± 0.41 0.23 ± 0.48 0.09 ± 0.32	0.70 ± 0.85 0.21 ± 0.29 0.09 ± 0.31	F_(2, 128)_ = 0.45; *p* = 0.637 F_(2, 128)_ = 1.05; *p* = 0.352 F_(2, 128)_ = 0.29; *p* = 0.747
**Calorie Intake in Race (kcal)**	Up to 3000 Up to 4000 Up to 6000	76% (100) 21% (27) 3% (4)	69% (20) 28% (8) 3% (1)	73% (32) 25% (11) 2% (1)	83% (48) 14% (8) 3% (2)	χ^2^_(4)_ = 3.13 *p* = 0.536
**Macronutrient Intake** **on Race Day**	Carbohydrate Protein Fat	69.6 ± 12.5 19.5 ± 11.9 12.5 ±10.8	71.2 ± 9.51 21.2 ± 16.6 14.5 ± 17.4	66.8 ± 15 19.9 ± 11.2 13.3 ± 9.04	70.8 ± 11.6 18.3 ± 9.41 10.9 ± 7.22	F_(2, 128)_ = 1.09; *p* = 0.338 F_(2, 128)_ = 0.49; *p* = 0.611 F_(2, 128)_ = 1.30; *p* = 0.275

Note. Results are presented in percentages (%), total numbers, and mean (SD). χ^2^ statistic calculated by Pearson’s chi-square test and F statistic by Kruskal–Wallis test. L–liters. kcal–kilocalories.

## Data Availability

If desired, inquirers will receive a brief summary of the results of the NURMI study. The datasets generated and/or analyzed during the current study and presented in this investigation are not publicly available. Requests to access the datasets can be directed to info@nurmi-study.com.

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
