# Peer review of "Dietary Habits and Race Day Strategies among Flexitarian, Vegetarian, and Vegan Recreational Endurance Runners: A Cross-Sectional Investigation from The NURMI Study (Step 2)"

_nutrients, 2024, doi:10.3390/nu16111647_

Round 1

Reviewer 1 Report

Comments and Suggestions for Authors

Thank you for submitting your manuscript. We understand the work put into this project. However, the reviewer cannot consider it for endorsement unless some major amendments are made. Please see our comments below:

Titel: Your title is very confusing. Is your project dealing with plant based diets or flexetarian diet? Please specify and consider adapting your title accordingly. 

General aspects: 

  • As for reasons for plant-based diets in athletes, the reviewers are missing possible anti-inflammatory claims of these dietary trends as motivating fact.  
  • Please have a native speaker or person fluent in English proof-read your manuscript. Especially, „Materials and Methods“ needs to be reworked language wise. 
  • If you analyze female and male individuals why do you not state their sociodemographic data gender-specifically (Table 1)? Averaging their data together does not seem appropriate. Please explain.

Line 22_23: Please adapt „changed … (to what)“.

Line 48: We would suggest removing „predominantly“ for this is an assumption rather than a fact. 

Line 72: Please refrain using terms like „archetypical“ or „utter“. Who defines such terms? This is not scientific and purely dependent on subjective and cultural perspectives of the reader. 

Line 73: Did they participate? If yo, „intended“ is the wrong term to use. 

Lines 221-224: Figure 3 remains inconclusive to the reviewer. Please explain in more detail. What is the purpose/need of this figure. 

Line 233: Your manuscript end with Table 2. Please include Table 3. 

Discussion:

The first 5 paragraphs of your discussion deal with reasons for a plant-based diet, whereas only the two last paragraphs actually deal with race day strategies. Please clarify the reasoning of your manuscript/project. Your hypothesis remains elusive right now. 

Comments on the Quality of English Language

The manuscript needs sufficient language editing my a native speaker or equivalent. 

Reviewer 2 Report

Comments and Suggestions for Authors

The aim of the present investigation was to compare race day strategies of recreational endurance runners following current dietary trends in sustainable eating (flexitarian, vegetarian, and vegan diets). It was assumed that recreational endurance runners following vegan diets require a complex race day dietary strategy compared to flexitarians or vegetarians to meet the demands of long-distance running performance and optimal recovery.

The manuscript is well structured and deals with a highly topical topic. However, I have some suggestions for authors.

The topic covered by the authors is highly topical, and the introductory paragraph deals with the topic well and provides a satisfactory theoretical background; however, to implement this section, I recommend authors consider the following article:

-        Villano ert al., Effects of vegetarian and vegan nutrition on body composition in competitive futsal athletes (2021) Progress in Nutrition, 23 (2).

Furthermore, although the purpose of the study is well described, it would be appropriate to better specify why the authors decided to carry out this study and also provide a research hypothesis.

The materials and methods section should be restructured by dividing it into paragraphs so as to make it easier to understand (Subejects, study design, statistical analyasis).

Authors must specify all acronyms used in Table 1 an 2.

The resolution of figure 2 should be increased.

The "Discussions" paragraph begins like this: This investigation aimed to analyze recreational endurance runners following different plant-based diets (flexitarian, vegetarian, vegan) and to compare their race day strategies.

However, this section should begin by highlighting the strengths of the study and then discuss the results based on what is already present in the literature.

Round 2

Reviewer 1 Report

Comments and Suggestions for Authors

Thank you for addressing all our comments.

Reading your manuscript again carefully, we understand better your intention regarding:

  • Combining both genders in one table
  • Language Editing: We appreciate the amendments that have been made. We agree that it is difficult to adapt the manuscript if detailed examples are missing. Furthermore, language and stylistic expression are often strongly influenced by the individual expression of the author, which is often misunderstood. The reviewers would therefore like to apologize for the ‘general criticism’ of the language aspects of this manuscript. 

To avoid possible subjective aspects, we have discussed your title with several colleagues. None were able to understand or estimate the content of your paper by the current title. Therefore we need to disagree with the authors and once more ask a revision of your title.

We agree that there is greater need in defining diets especially if they include various aspects of different diets such as the flexitarian diet. 

However, we fail to understand the need of discussing race day strategies of plant-based diets AND the gap of information/definition regarding flexetarian diet in ONE manuscript. We believe combining both aspects in one scientific project makes this paper unnecessary difficult to understand and follow. 

Referring to your answer No12:

„We have further clarified the aim of the present investigation, which was (1) to analyze the current dietary subgroups (flexitarian, vegetarian, vegan) and (2) to compare their race day strategies.“

From our point of view, this is unfortunately neither reflected by the title nor the current abstract. Also, we would like to add, citing your aim of page 10:

„The NURMI study was established as the first of it’s kind to compare the performances of long-distance runners around the world following omnivorous, vegetarian, and vegan diets.“

Again, performance has not yet been part of this paper. 

We need to conclude that the paper itself - although reflecting hard work - is hard to read and unclear to estimate its clear goals. We believe if you consider defining this more clearly and following this red-line, the manuscript would greatly benefit from it. 

Round 3

Reviewer 1 Report

Comments and Suggestions for Authors

Thank you for addressing all our comments. The criticized aspects have been explained or adapted. From our point of view your title reflects the storyline of your manuscript much better. 

The line of your overall scientific work with the NURMI study is much clearer now. Thank you again.